# Clinical Characteristics of Patients with Ossification of the Posterior Longitudinal Ligament and a High OP Index: A Multicenter Cross-Sectional Study (JOSL Study)

**DOI:** 10.3390/jcm11133694

**Published:** 2022-06-27

**Authors:** Takashi Hirai, Toshitaka Yoshii, Jun Hashimoto, Shuta Ushio, Kanji Mori, Satoshi Maki, Keiichi Katsumi, Narihito Nagoshi, Kazuhiro Takeuchi, Takeo Furuya, Kei Watanabe, Norihiro Nishida, Soraya Nishimura, Kota Watanabe, Takashi Kaito, Satoshi Kato, Katsuya Nagashima, Masao Koda, Hiroaki Nakashima, Shiro Imagama, Kazuma Murata, Yuji Matsuoka, Kanichiro Wada, Atsushi Kimura, Tetsuro Ohba, Hiroyuki Katoh, Masahiko Watanabe, Yukihiro Matsuyama, Hiroshi Ozawa, Hirotaka Haro, Katsushi Takeshita, Morio Matsumoto, Masaya Nakamura, Satoru Egawa, Yu Matsukura, Hiroyuki Inose, Atsushi Okawa, Masashi Yamazaki, Yoshiharu Kawaguchi

**Affiliations:** 1Department of Orthopedic Surgery, Tokyo Medical and Dental University, Bunkyo, Tokyo 113-8519, Japan; yoshii.orth@tmd.ac.jp (T.Y.); 0123456789jun@gmail.com (J.H.); ushiorth20@gmail.com (S.U.); egawa.orth@tmd.ac.jp (S.E.); matsukura.orth@tmd.ac.jp (Y.M.); inose.orth@tmd.ac.jp (H.I.); okawa.orth@tmd.ac.jp (A.O.); 2Department of Orthopaedic Surgery, Shiga University of Medical Science, Ōtsu 520-2192, Japan; kanchi@belle.shiga-med.ac.jp; 3Department of Orthopedic Surgery, Chiba University Graduate School of Medicine, Chiba 260-8670, Japan; satoshi.maki@chiba-u.jp (S.M.); takeo251274@yahoo.co.jp (T.F.); 4Department of Orthopedic Surgery, Niigata University Medical and Dental General Hospital, Niigata 951-8520, Japan; kkatsu_os@yahoo.co.jp (K.K.); keiwatanabe_39jp@live.jp (K.W.); 5Department of Orthopedic Surgery, Keio University School of Medicine, Shinjuku, Tokyo 160-8582, Japan; nagoshiation@gmail.com (N.N.); soraya.nishimura@gmail.com (S.N.); watakota@gmail.com (K.W.); morio@a5.keio.jp (M.M.); masa@a8.keio.jp (M.N.); 6Department of Orthopedic Surgery, National Hospital Organization Okayama Medical Center, Okayama 701-1192, Japan; takeuchi.kazuhiro.qr@mail.hosp.go.jp; 7Department of Orthopedic Surgery, Yamaguchi University Graduate School of Medicine, Ube 755-8505, Japan; nishida3@yamaguchi-u.ac.jp; 8Department of Orthopaedic Surgery, Osaka University Graduate School of Medicine, Suita 565-0871, Japan; takashikaito@gmail.com; 9Department of Orthopedic Surgery, Graduate School of Medical Sciences, Kanazawa University, Kanazawa 920-8641, Japan; skato323@gmail.com; 10Department of Orthopedic Surgery, Faculty of Medicine, University of Tsukuba, Tsukuba 305-8577, Japan; katsu_n103@yahoo.co.jp (K.N.); masaokod@gmail.com (M.K.); masashiy@md.tsukuba.ac.jp (M.Y.); 11Department of Orthopedic Surgery, Graduate School of Medicine, Nagoya University, Nagoya 464-8601, Japan; hirospine@med.nagoya-u.ac.jp (H.N.); imagama@med.nagoya-u.ac.jp (S.I.); 12Department of Orthopedic Surgery, Tokyo Medical University, Shinjuku, Tokyo 160-8402, Japan; kaz.mur26@gmail.com (K.M.); yuji_kazu77@yahoo.co.jp (Y.M.); 13Department of Orthopedic Surgery, Hirosaki University Graduate School of Medicine, Hirosaki 036-8562, Japan; wadak39@hirosaki-u.ac.jp; 14Department of Orthopedics, Jichi Medical University, Shimotsuke 329-0498, Japan; akimura@jichi.ac.jp (A.K.); dtstake@gmail.com (K.T.); 15Department of Orthopedic Surgery, University of Yamanashi, Chuo 409-3898, Japan; tooba@yamanashi.ac.jp (T.O.); haro@yamanashi.ac.jp (H.H.); 16Department of Orthopedic Surgery, Surgical Science, Tokai University School of Medicine, Isehara 259-1193, Japan; hero@tokai-u.jp (H.K.); masahiko@is.icc.u-tokai.ac.jp (M.W.); 17Department of Orthopedic Surgery, Hamamatsu University School of Medicine, Hamamatsu 431-3125, Japan; spine-yu@hama-med.ac.jp; 18Department of Orthopaedic Surgery, Tohoku Medical and Pharmaceutical University, Sendai 983-8536, Japan; hozawa@tohoku-mpu.ac.jp; 19Department of Orthopedic Surgery, Faculty of Medicine, University of Toyama, Toyama 930-8555, Japan; zenji@med.u-toyama.ac.jp

**Keywords:** ossification of posterior longitudinal ligament, prospective multi-institutional study, pain, patient-reported outcomes, OP index

## Abstract

Background: The purpose of this study was to clarify the clinical features of ossification of the posterior longitudinal ligament (OPLL) and extreme ossification at multiple sites. Methods: This prospective study involved patients with a diagnosis of cervical OPLL at 16 institutions in Japan. Patient-reported outcome measures, including responses on the Japanese Orthopaedic Association (JOA) Cervical Myelopathy Evaluation Questionnaire (JOA-CMEQ), JOA Back Pain Evaluation Questionnaire (JOA-BPEQ), and visual analog scale pain score, were collected to investigate clinical status. In each patient, the sum of the levels at which OPLL was located (OP index) was evaluated on whole-spine computed tomography, along with ossification of other spinal ligaments including the anterior longitudinal ligament (OALL), ligament flavum (OLF), supra- and intraspinous ligaments (SSL), and diffuse idiopathic skeletal hyperostosis (DISH). The distribution of OP index values in the study population was investigated, and the clinical and radiologic characteristics of patients in the top 10% were assessed. Results: In total, 236 patients (163 male, 73 female; mean age 63.5 years) were enrolled. Twenty-five patients with OP index ≥ 17 were categorized into a high OP index group and the remainder into a moderate/low OP index group. There were significantly more women in the high OP index group. Patients in the high OP index group also had significantly poorer scores for lower extremity function and quality of life on the JOA-CMEQ and in each domain but not for body pain on the JOA-BPEQ compared with those in the moderate/low OP index group. Patients in the high OP index group had more OALL in the cervical spine and more OLF and SSL in the thoracic spine. The prevalence of DISH was also significantly higher in the high OP index group. In the high OP index group, interestingly, OPLL was likely to be present adjacent to DISH in the cervicothoracic and thoracolumbar spine, especially in men, and often coexisted with DISH in the thoracic spine in women. Conclusion: This prospective cohort registry study is the first to demonstrate the clinical and radiologic features of patients with OPLL and a high OP index. In this study, patients with a high OP index had poorer physical function in the lumbar spine and lower extremities and were also predisposed to extreme ossification of spinal ligaments other than the OPLL.

## 1. Introduction

Ossification of the spinal ligaments is the term used to describe heterotopic ossified lesions that can occur throughout the spine. Ossification of the posterior longitudinal ligament (OPLL) has been recognized as a common cause of neurologic disorders, including myelopathy, radiculopathy, and/or amyotrophy. Epidemiological studies have shown that cervical OPLL has a male predominance of 2:1 to 3:1 [1], whereas thoracic OPLL has a female predominance of 2:1 to 3:1 [2,3]. Moreover, a recent genome-wide association study identified six candidate genes for the pathogenesis of OPLL [4]. Therefore, the onset and expansion of ossified lesions may be closely associated with sex hormones and genetic background.

Various groups have reported on OPLL, including its epidemiology [1,2,3], surgical treatment [5,6,7], and radiological features [8,9,10,11,12,13,14,15,16]. Computed tomography (CT) has been widely used to evaluate bone structure and the distribution of ossification in the whole spine. It has been documented that the onset and progression of neurological symptoms are associated with the degree of canal narrowing caused by OPLL and segmental mobility at the level where OPLL is present [5]. Patients with OPLL and excessive spinal cord compression due to OPLL often have multiple lesions throughout the spine [9]. Given that thoracic OPLL sometimes leads to a concurrent spinal cord disorder with cervical ossification, it is important to evaluate ossified lesions and diagnose the level at which compression caused by ossification results in neurological deterioration.

Although the distribution of ossified lesions has been well investigated in other studies [10,11], there has been no research on the clinical characteristics of patients with OPLL in whom ossification occurs in a large number of segments. We have defined an OP index which reflects the number of segments where OPLL is present and evaluated the predisposition to ossification in individual patients with OPLL [12,13,14,15,16]. In this study, we investigated the clinical features of patients with a large number of ossifications and a high OP index and characteristics of bony bridging lesions of both DISH and OPLL in patients with a high OP index using data from a nationwide patient registry started by the Japanese Multicenter Research Organization for Ossification of the Spinal Ligament (JOSL), which was established by the Japanese Ministry of Health, Labour and Welfare.

## 2. Materials and Methods

This study had a multi-institutional prospective cross-sectional design and analyzed data from patients who visited 16 JOSL member institutions between September 2015 and December 2017. The patients in this study largely overlapped with those in our previous study, but here our analysis focused on the characteristics of only those patients with a high OP index. The inclusion criteria, clinical evaluation, and radiographic evaluation were described previously [9]. The inclusion criteria were age ≥ 20 years; cervical OPLL diagnosed based on radiographic findings; symptoms such as neck pain and upper and/or lower extremity numbness regardless of the need for surgery and presence of clumsiness and gait disturbance; and whole-spine CT scans available for localizing ossified lesions. Patients with a history of cervical spine surgery for OPLL were excluded. The study was approved by the institutional review board of each participating institution and conducted in accordance with the relevant guidelines and regulations.

The baseline demographic and clinical data listed in Table 1 and Table 2 were collected for all patients, and included evaluation of the cervical Japanese Orthopaedic Association (JOA) score for functional assessment in patients with cervical myelopathy [17]; the JOA Cervical Myelopathy Evaluation Questionnaire (JOA-CMEQ) for assessment of quality of life and functional assessment of the cervical spine, upper and lower extremities, and bladder and [18]; and the JOA Back Pain Evaluation Questionnaire (JOA-BPEQ) lumbar spine function, social dysfunction, mentality, locomotive function, and body pain [18]. A visual analog scale (VAS) was used to assess severity of pain or stiffness in the neck or shoulders, pain or numbness in the arms or hands, and low back pain.

Radiological evaluation was performed as in our previous study [9]. Briefly, whole-spine CT images were collected for each patient, including the cervical, thoracic, and lumbosacral spine. The incidence of OPLL was examined on mid-sagittal CT images in the cervical spine (the clivus to C7) and in the thoracic and lumbosacral regions from T1 to S1. As in previous studies [9,13,14,15,16], images were independently evaluated by senior spine surgeons (S.U., K.M., S.M., K.K., N.N., and K.T.) blinded to clinical outcomes. The OP index was defined as the number of levels with OPLL in the whole spine [9,16] and was also calculated for each patient.

The patients were then divided into a high OP index group (top 10% of OP index values) and a moderate/low OP index group. As in previous reports [13,14,19], the numbers of ossified spinal ligaments other than OPLL were also evaluated, namely, the anterior longitudinal ligament (OALL), ligament flavum (OLF), supra- and interspinous ligaments, and nuchal ligament. In addition, we evaluated the extent to which OPLL occupied the spinal canal in the cervical region and classified the most compressed segment according to the canal narrowing ratio (CNR) [7] as follows: Grade 1, 0–25%; Grade 2, 26–50%; Grade 3, 51–75%; Grade 4, >75%. OPLL was assessed as diffuse idiopathic skeletal hyperostosis (DISH) if it completely bridged at least 4 contiguous adjacent vertebral bodies according to the criteria reported by Nishimura et al. [20]. The distributions of OPLL and DISH in the whole spine were noted for each case in the high OP index group.

The two groups were compared using the unpaired *t*-test, Mann-Whitney *U* test, and chi-squared test as appropriate. The data are presented as the mean ± standard deviation. All statistical analyses were performed using SPSS for Windows (ver. 22.0; IBM Corp., Armonk, NY, USA). A *p*-value of less than 0.05 was considered statistically significant.

## 3. Results

### 3.1. Distribution of OP-Index Values in Patients with OPLL

A single peak wave was found in the distribution of OP index values in the study population. The patients with the top 10% OP index value (*n* = 25) had at least 17 ossified lesions (Figure 1). These cases were defined as the high OP index group, and the remaining patients (*n* = 211) in whom the OP index was ≤16 were defined as the moderate/low OP index group.

### 3.2. Demographic Characteristics in the High and Moderate/Low OP Index Groups

The patient demographics are shown in Table 1. Mean age was 59.0 years in the high OP index group and 64.5 years in the moderate/low OP index group. Patients in the high OP index group were relatively younger, although the difference was not statistically significant. There were significantly fewer men in the high OP index group; however, there was no significant difference between the groups in body mass index, prevalence of diabetes mellitus, or cervical JOA score.

### 3.3. Deterioration of Lumbar Spine and Lower Extremity Function in the High OP-Index Group

Patient-reported evaluations, including JOA-CMEQ, JOA-BPEQ, and VAS scores, were assessed to determine whether neurological symptoms were associated with the severity of ossified lesions in the high OP index group (Table 2). There was no significant difference between the two groups in the prevalence of pain. On the JOA-CMEQ, patients in high OP index group had significantly poorer lower extremity function and quality of life scores compared with those in the moderate/low OP index group; however, there was no significant difference in cervical spine, upper extremity, or bladder function. There was also a significant difference in the score for each domain of the JOA-BPEQ, with the exception of body pain. VAS scores were significantly higher for numbness below the chest, low back pain, lower extremity numbness, and lower extremity pain in the high OP index group but no difference was noted in neck pain, numbness in the upper extremities, or chest constriction.

### 3.4. Higher Prevalence of Ossified Thoracic Lesions in Patients with a High OP Index

The distributions of OPLL and ossification of other spinal ligaments were compared between the two groups (Figure 2A–E). The number of ossifications, especially in the thoracic spine, was markedly higher in the high OP index group. (Figure 2A). Patients in the high OP index group also had more OALL in the cervical spine (Figure 2B) and more OLF (Figure 2C) and ossification of the supra- and interspinous ligaments (Figure 2D) in the thoracic spine. DISH was significantly more prevalent in the high OP index group than in the moderate/low OP index group (Figure 2E). The high OP index group had a significantly higher CNR compared with the moderate/low OP index group (Table 3).

### 3.5. Distribution of OPLL and DISH in the High OP Index Group

The distribution of ossification in each patient in the high OP index group was investigated in detail. DISH was found in 9 men (100%) and 10 women (62.5%), suggesting that DISH is more prevalent in men (*p* = 0.03). OPLL was likely to be present adjacent to DISH in the cervicothoracic and thoracolumbar spine, especially in men, and often coexisted with DISH in the thoracic spine in women (Figure 3). Interestingly, patients with a relatively high OP index in the moderate/low OP group also showed similar differences according to sex.

## 4. Discussion

Ossification of the spinal ligaments is more prevalent in East Asian countries and has been recognized as heterotopic bone formation in the spine. The prevalence of OPLL in the Japanese population has been reported to range from 1.9% to 6.3% [1,8,21,22]. In particular, OPLL can be a common cause of myelopathy. Liao et al. reviewed three-dimensional CT scans for 7210 patients with degenerative cervical myelopathy and showed that the prevalence of OPLL was as high as 18% [23]. However, Fujimori et al. reviewed the registry data of OPLL patients and demonstrated that the prevalence of symptomatic OPLL was less than 1% in those with radiographically detected OPLL [21], demonstrating that most cases with radiographically detectable OPLL are asymptomatic. Thus, the mechanisms of onset and progression of myelopathy caused by OPLL is not fully understood.

A previous report [12] showed that the amount of OPLL (i.e., the OP index) was significantly correlated with the cervical OP index, female sex, and obesity. We also demonstrated that male sex was significantly less prevalent in the high OP index group. Furthermore, we have previously shown that the greater the OP index in patients with OPLL, the greater canal narrowing ratio [24]. Patients with a higher OP index may also be more likely to develop spinal cord compression and myelopathy. In the present study, the cervical JOA score was lower, albeit not significantly, in the high OP index group than in the moderate/low OP index group. In patients with a high OP index, ossified lesions can compress the spinal cord, leading to symptoms of myelopathy. Therefore, patients who are asymptomatic but have a high OP index should undergo long-term follow-up because neurological deterioration may occur.

In our study, evaluation of the patient-reported JOA-CMEQ measures showed that lower extremity function and quality of life were significantly worse in the high OP index group without any significant difference in cervical spine function, upper extremity function, bladder function, quality of life, or body pain. It has been reported that the OP index in the cervical spine is significantly associated with lower extremity motor function [24]. This indicates that the posterior column can be impaired by ossification in patients with a higher OP index. Haddas et al. compared the relationship between spatiotemporal parameters and the kinematics of the spine and lower extremities during the gait cycle between patients with myelopathy and healthy volunteers. Interestingly, they found that patients with myelopathy have significantly increased anterior pelvic tilt and lumbar lordosis as well as less cervical lordosis and head flexion, and consequently had less range of motion at the knee during the gait cycle [25]. Therefore, gait disturbance caused by myelopathy may not only influence lumbar spinal function but also cause locomotive dysfunction in the lower limbs.

Clinically, spinal disorders can result from spinal cord compression caused by OPLL or ossification of other spinal ligaments. OLF often coexists with OPLL and results in onset of neurological dysfunction at the cervicothoracic and thoracolumbar junctions [9,24]. Lee et al. reviewed 101 patients with symptomatic OLF and 102 patients with asymptomatic OLF that was discovered incidentally during investigation of thoracic or lumbar compression fractures to identify radiologic characteristics of symptomatic OLF and showed that the fused and tuberous types of OLF mostly lead to myelopathy [26]. They also identified an ossified cross-sectional area of the spinal canal of 33% as the cutoff value for a diagnosis of thoracic myelopathy caused by OLF. We found that the amount of OLF was greater in the high OP index group. Therefore, it is important for spine surgeons to evaluate not only OPLL but also OLF in the whole spine.

The present study also showed that patient-reported outcome measures related to the lumbar spine on the JOA-BPEQ were significantly worse in the high OP index group than in the moderate/low OP index group. In the high OP index group, ossification of multiple spinal ligaments, including OPLL, OLF, and SSL, was observed in the thoracic spine but not in the lumbar spine (Figure 2). Moreover, the prevalence of DISH was significantly higher in patients with a high OP index. Yamada et al. reviewed the CT myelograms for 61 patients who underwent surgery for symptomatic thoracic OLF and reported lumbar spinal canal stenosis as an incidental finding in up to 75% of cases [27]. Furthermore, a systematic review that included 524 patients with thoracic spinal stenosis found that the prevalence of lumbar canal stenosis was 35.9% [28]. These findings suggest that ankylosis in the thoracic spine might cause a long lever arm and then create excess loading on the lumbar spine. Despite the structural changes in the whole spine, patients with myelopathy due to ossification of spinal ligaments often present with pain, numbness, or impaired lower limb motor function. These symptoms likely increase the burden on the lumbar spine where mobile segments are preserved.

This study is the first to investigate the relationship between distribution of OPLL and DISH in patients with a predisposition to ossification in detail and found an extremely high prevalence of DISH in a high OP index group. In a retrospective case series by our group [20], DISH was found in 48.7% of patients with cervical OPLL and was basically distributed in the middle thoracic spine in younger cases but could extend to more proximal and/or distal levels in the whole spine in older cases. Although the mean age was relatively younger in the high OP index group, the prevalence of DISH was significantly higher than in the moderate/low group (76% vs. 43.6%). Our present findings indicate that OPLL and OALL can occur at the same time and extend even in younger patients who are predisposed to ossification. Interestingly, in the high OP index group, all of the men had DISH while 6 (37.5%) of the women did not. We also found that OPLL was likely to be present adjacent to DISH in the cervicothoracic and thoracolumbar spine in men, and that OPLL often coexisted with DISH in the thoracic spine in women. This evidence suggests that structural characteristics of ossification might be different between genders. Various factors have been shown to regulate the onset and extension of ossification and various sex-related mechanisms have been suggested [11,29,30,31]. Serum estrogen level has been shown to correlate with the onset and extension of ossified lesions, and estrogen is thought to stimulate osteoblast-like cells with the aid of trophic factors [11,30,31]. According to Ikeda et al., serum leptin and insulin levels were significantly higher in women with OPLL and there is a positive correlation between serum leptin level and both the affected levels with OPLL and the extent of ossification in the thoracic and/or lumbar spine [32]. These metabolic factors could affect the onset and development of OPLL in the whole spine over and above a sex-related difference in the prevalence of ossification.

Rheumatologic pathologies have also been shown to be associated with ossification predisposition. Notably, ankylosing spondylitis (AS), which is also known as “bamboo spine” [33] and consists of annulus fibrosus ossification and adjacent vertebral body bridging anteriorly and laterally, is characterized by chronic inflammatory disease and presence of the HLA-B27 antigen [33,34]. The Modified New York Criteria [35] defines AS based on radiographic features, including sacroiliitis grade ≥ 2 bilaterally or grade 3–4 unilaterally. Although vertebral ossification bridging can be found in DISH and/or OPLL [36,37] with morphology similar to that of AS, bone formation in patients with ossification differs from that in AS patients on a basic level. The bridging formation in ossification patients often has a candle-wax pattern. However, it is not presently clear how to recognize ossification of the spinal ligaments and AS. Nguyen et al. reviewed 104 DISH patients who also had OPLL and divided these patients into three groups according to bridging pattern [38]. They showed that patients with a flat bridging pattern, similar to the morphology of AS, had higher high-sensitivity CRP levels compared with patients with a jagged bridging pattern. This evidence suggests that flat vertebral bridging, which is often found in AS patients, might be caused by inflammatory pathogenesis rather than a degenerative process. However, further investigation is required to definitively determine the pathology.

This study has some limitations that should be noted. First, it had a cross-sectional cohort design, investigated a specific disease, and was not population-based. Second, the longitudinal design precludes any discussion of causality. Third, the JOA-CMEQ and JOA-BPEQ could not evaluate pain states in detail. These clinical questions should be addressed in future work by conducting population-based studies and adjusting for confounding factors for each spinal ligament. These limitations notwithstanding, this work contributes to understanding the clinical features of DISH in patients with cervical OPLL.

## 5. Conclusions

This prospective study used data from a nationwide cohort registry to investigate the clinical and radiologic features of patients with OPLL and a high OP index. We demonstrated that patients with a high OP index had poorer physical function in the lumbar spine and lower extremities and a predisposition to extreme ossification, including in spinal ligaments other than the OPLL.

## Figures and Tables

**Figure 1 jcm-11-03694-f001:**
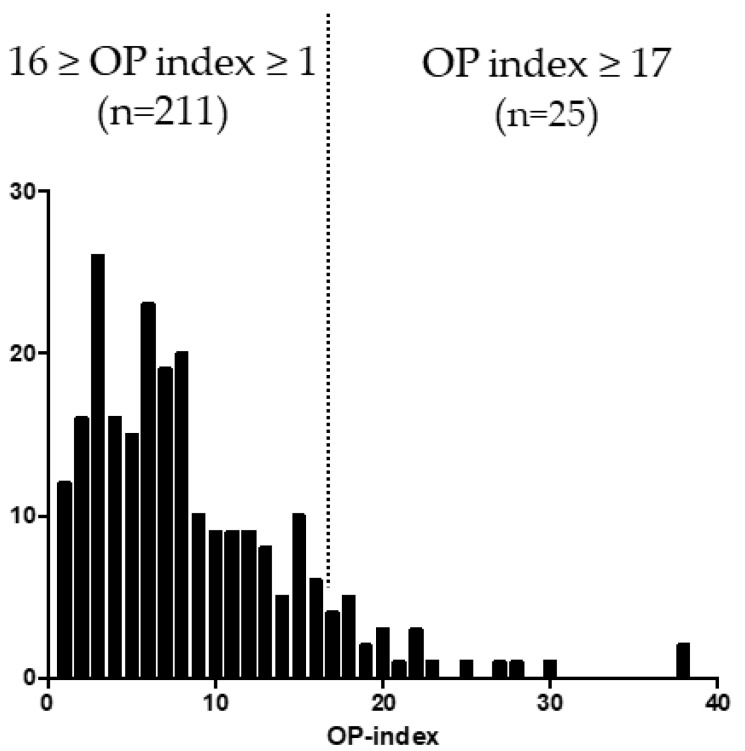
The distribution of OP index values.

**Figure 2 jcm-11-03694-f002:**
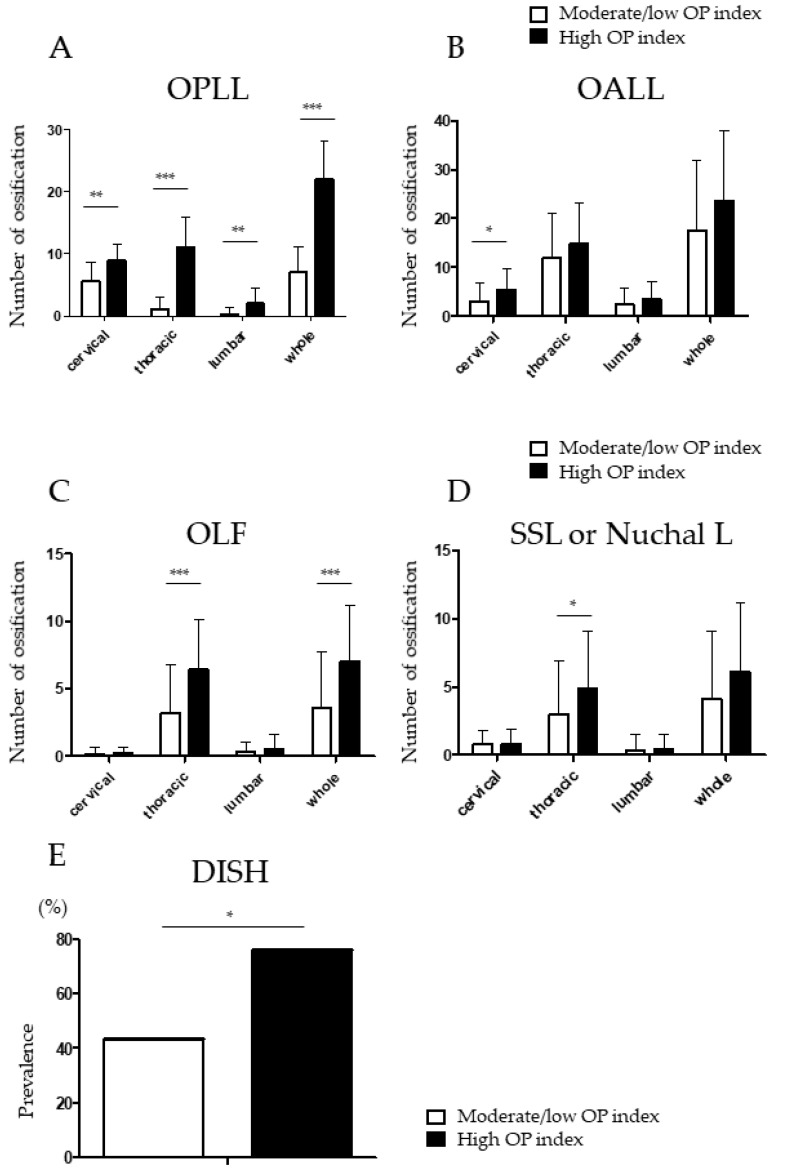
Radiologic findings, including OPLL (**A**), OALL (**B**), OLF (**C**), SSL or nuchal ligament (**D**), and DISH (**E**) in the high OP-index group and moderate/low OP-index group. DISH, diffuse idiopathic skeletal hyperostosis; OALL, ossification of the anterior longitudinal ligament; OLF, ossification of the ligamentum flavum; OPLL, ossification of the posterior longitudinal ligament; SSL, supra- and interspinous ligaments * *p* < *0*.05, ** *p* < *0*.01, *** *p* < *0*.001.

**Figure 3 jcm-11-03694-f003:**
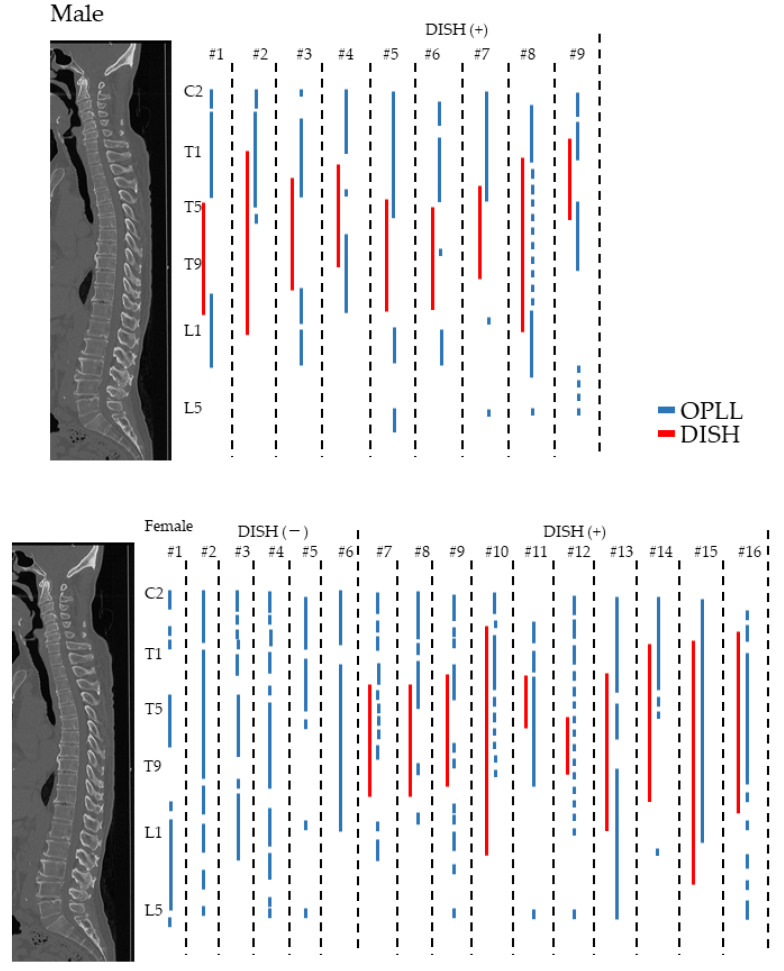
The distribution of OPLL and DISH in men and women in the high OP index group. DISH, diffuse idiopathic skeletal hyperostosis; OPLL, ossification of the posterior longitudinal ligament.

**Table 1 jcm-11-03694-t001:** Demographic and clinical characteristics of patients with ossification of the posterior longitudinal ligament according to whether they had a high or moderate/low OP index.

	High OP Index (*n* = 25)	Moderate/Low OP Index (*n* = 211)	*p*-Value
Age (years)	59.0 ± 14.0	64.5 ± 12.0	0.07
Male sex (%)	32.0	73.5	<0.001 ***
Body mass index	27.4 ± 5.8	25.7 ± 4.3	0.17
Diabetes mellitus (%)	28.0	24.6	0.82
Cervical JOA score	11.5 (6–17)	12.4 (−2, 17)	0.12

Data are expressed as the mean ± standard deviation. JOA, Japanese Orthopaedic Association. *** *p* < 0.001.

**Table 2 jcm-11-03694-t002:** Clinical characteristics in patients with ossification of the posterior longitudinal ligament according to whether the OP index was high or moderate/low.

	High OP Index (*n* = 25)	Moderate/Low OP Index (*n* = 211)	*p*-Value
Prevalence of symptoms (%)			
Neck pain	60.0	60.2	0.98
Back pain	32.0	28.0	0.67
Low back pain	68.0	52.6	0.14
JOA-CMEQ score			
Cervical spine function	59.8 ± 33.8	66.6 ± 27.9	0.34
Upper extremity function	74.5 ± 21.6	80.8 ± 21.5	0.18
Lower extremity function	46.1 ± 35.7	68.3 ± 29.3	0.006 **
Bladder function	73.0 ± 20.2	74.7 ± 22.3	0.69
Quality of life	40.4 ± 20.2	51.0 ± 19.6	0.02 *
JOA-BPEQ score			
Lumbar spine function	54.0 ± 29.3	69.7 ± 31.6	0.02 *
Social dysfunction	43.4 ± 32.8	57.9 ± 28.7	0.04 *
Mentality	40.8 ± 18.6	50.1 ± 19.9	0.03 *
Locomotive function	45.6 ± 37.1	66.4 ± 34.5	0.01 *
Body pain	58.2 ± 39.1	72.3 ± 32.7	0.09
VAS score			
Neck pain	40.2 ± 31.3	38.6 ± 31.2	0.81
Upper extremity numbness	47.5 ± 34.5	44.8 ± 33.2	0.71
Chest constriction	13.0 ± 27.9	9.9 ± 21.0	0.60
Numbness below the chest	60.1 ± 28.9	34.2 ± 33.6	<0.001 ***
Low back pain	44.1 ± 32.9	25.7 ± 27.9	0.01 *
Lower extremity numbness	54.4 ± 34.9	28.1 ± 32.6	<0.01 **
Lower extremity pain	37.8 ± 33.6	21.4 ± 29.4	0.03 *

Data are expressed as the mean ± standard deviation. BPEQ, Back Pain Evaluation Questionnaire; CMEQ, Cervical Myelopathy Evaluation Questionnaire; JOA, Japanese Orthopaedic Association; VAS, visual analog scale. * *p* < 0.05, ** *p* < 0.01, *** *p* < 0.001.

**Table 3 jcm-11-03694-t003:** CNR grade according to high or moderate/low OP index values.

CNR Grade	High OP Index (*n* = 25)	Moderate/Low OP Index (*n* = 211)	*p*-Value
Grade 1	2 (8%)	70 (33.2%)	
2	8 (32%)	71 (33.6%)	
3	12 (48%)	56 (26.5%)	
4	3 (12%)	14 (6.6%)	
			0.03 *

CNR, canal narrowing ratio. * *p* < 0.05.

## Data Availability

The data generated and analyzed in this study are available from the corresponding author upon reasonable request.

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
