# Peer review of "Clinical Characteristics of Patients with Ossification of the Posterior Longitudinal Ligament and a High OP Index: A Multicenter Cross-Sectional Study (JOSL Study)"

_jcm, 2022, doi:10.3390/jcm11133694_

Round 1

Reviewer 1 Report

The work is of considerable interest and can be accepted in my opinion, however, since it is a journal of express medical interest, the authors, with the availability of such a large multicentric series, have not considered a fundamental aspect that is the etiopathogenesis of ossification of the posterior longitudinal ligament (OPLL), anterior longitudinal ligament (OALL), ligament flavum (OLF), supra- and intraspinous ligaments (SSL), and diffuse idiopathic skeletal hyperostosis (DISH).

In the discussion page 12, line 2013 and 278-284 the prevalence of OPLL in the Japanese population has been reported and estrogen level has been shown to correlate with the onset and extension of ossified lesions.

However, the rheumatological and autoimmune pathologies, that are known causes of the pathology analyzed in this work, are never mentioned.

Author Response

The work is of considerable interest and can be accepted in my opinion, however, since it is a journal of express medical interest, the authors, with the availability of such a large multicentric series, have not considered a fundamental aspect that is the etiopathogenesis of ossification of the posterior longitudinal ligament (OPLL), anterior longitudinal ligament (OALL), ligament flavum (OLF), supra- and intraspinous ligaments (SSL), and diffuse idiopathic skeletal hyperostosis (DISH).

In the discussion page 12, line 2013 and 278-284 the prevalence of OPLL in the Japanese population has been reported and estrogen level has been shown to correlate with the onset and extension of ossified lesions. However, the rheumatological and autoimmune pathologies, that are known causes of the pathology analyzed in this work, are never mentioned.

Response: Thank you very much for reviewing our paper. Based on this comment, we have added the following text to the Discussion section.

“Rheumatologic pathologies have also been shown to be associated with ossification predisposition. Notably, ankylosing spondylitis (AS), which is also known as “bamboo spine” [33] and consists of annulus fibrosus ossification and adjacent vertebral body bridging anteriorly and laterally, is characterized by chronic inflammatory disease and the presence of the HLA-B27 antigen [33,34]. The Modified New York Criteria [35] defines AS based on radiographic features, including sacroiliitis grade ≥2 bilaterally or grade 3–4 unilaterally. Although vertebral ossification bridging can be found in DISH and/or OPLL [36,37] with morphology similar to that of AS, bone formation in patients with ossification is differs from that in AS patients on a basic level. The bridging formation in ossification patients often has a candle-wax pattern. However, it is not presently clear how to recognize ossification of the spinal ligaments and AS. Nguyen et al. reviewed 104 DISH patients who also had OPLL and divided these patients into three groups according to bridging pattern [38]. They showed that patients with a flat bridging pattern, similar to the morphology of AS, had higher high-sensitivity CRP levels compared with patients with a jagged bridging pattern. This evidence suggests that flat vertebral bridging, which is often found in AS patients, might be caused by inflammatory pathogenesis rather than a degenerative process. However, further investigation is required to definitively determine the pathology.” (page 9, lines 292–307)

Reviewer 2 Report

Hirai et al reported the feature of OPLL patient with their prospective database. They especially focused on OP-index and assessed the difference between high and low-moderate OP-index groups.

The biggest concern for me is that the novelty and meaning of this study. Even if considering these analyses are from the dataset of multi-center prospective study, provided knowledge may be limited. In the discussion section, the authors stated that “this work contributes to understanding the clinical features of DISH in patients with cervical OPLL.”. However, I cannot agree with this idea considering the results provided with this article. Further, several concerns and questions exist as below.

-Why the authors defined top 10 % of dataset as high OP-index group? How about 2.5 % or 5 %? These numbers are more frequently used for data analysis. For addressing the correlation between the OP-index and symptoms, regression analysis would provide us more accurate information. The degree of canal stenosis also should be included for multiple regression analysis (Of course, other factors should be included as well).

-The authors stated that OPLL was located adjacent to DISH. How about the difference between the high and moderate-low index groups?

-Patients with high OP-index showed sever symptoms comparing to those with moderate-low OP-index. Then, how about the degree of canal stenosis? Are there any correlation between OP-index and canal stenosis?

Minor comments below

-Why did the authors used a term of “OP-index”? The original report (Kawaguchi et al, Spine, 2013) in which this analysis was used reported this method as “OS-index”.

-The authors stated that “The OP-index value was markedly higher, especially in the thoracic spine, in the high OP index group.” However, it looks like that there is no data which suggests this idea. P value cannot be the evidence for this statement.

-In the discussion section, the authors stated that cervical OP-index, female sex, and obesity were significantly correlated with OP-index. Which data indicated the significant correlation between cervical OP-index or obesity with OP-index? I cannot find them.

Author Response

The biggest concern for me is that the novelty and meaning of this study. Even if considering these analyses are from the dataset of multi-center prospective study, provided knowledge may be limited. In the discussion section, the authors stated that “this work contributes to understanding the clinical features of DISH in patients with cervical OPLL.”. However, I cannot agree with this idea considering the results provided with this article. Further, several concerns and questions exist as below.

-Why the authors defined top 10 % of dataset as high OP-index group? How about 2.5 % or 5 %? These numbers are more frequently used for data analysis. For addressing the correlation between the OP-index and symptoms, regression analysis would provide us more accurate information. The degree of canal stenosis also should be included for multiple regression analysis (Of course, other factors should be included as well).

Response: Thank you very much for reviewing our paper. Because this study involved 236 patients, we could not assess the differences between the top 2.5% or 5% arm and another arm because it would affect the quality of the statistical analysis. Therefore, we chose the top 10% population in this study.

As pointed out, we performed a multivariate regression analysis to show the correlation between the OP-index and symptoms in a previous study.[1] Indeed, there were no significant correlations between the OP index and clinical symptoms except for neurological status (JOA score). Furthermore, we attempted to perform multivariate analysis using clinical factors combined with radiologic factors, including the degree of ossification of other spinal ligaments and the canal narrowing ratio. Because these radiologic-independent factors confounded with each other, we could not obtain a good result from multivariate analysis. Therefore, we divided these 236 patients into two groups to clarify the clinical features of patients predisposed to high ossification.

[1] Hirai, T.; Yoshii, T.; Ushio, S.; Hashimoto, J.; Mori, K.; Maki, S.; Katsumi, K.; Nagoshi, N.; Takeuchi, K.; Furuya, T.; et al. Associations between Clinical Symptoms and Degree of Ossification in Patients with Cervical Ossification of the Posterior Longitudinal Ligament: A Prospective Multi-Institutional Cross-Sectional Study. J Clin Med 2020, 9, doi:10.3390/jcm9124055.

-The authors stated that OPLL was located adjacent to DISH. How about the difference between the high and moderate-low index groups?

Response: Thank you for your question. We examined the distribution of OPLL and DISH and found that although patients with a low OP index were less likely to have DISH, patients with a relatively high OP index in the moderate-low group had a tendency similar to the high group. To address this, we have added the following text: “Interestingly, patients with a relatively high OP-index in the moderate/low OP group also showed similar differences according to sex.”. (page 7, lines 212–213)

-Patients with high OP-index showed sever symptoms comparing to those with moderate-low OP-index. Then, how about the degree of canal stenosis? Are there any correlation between OP-index and canal stenosis?

Response: Based on this comment, we have also measured the degree of canal stenosis (canal narrowing rate; CNR) in each patient and investigated whether there was a difference in CNR between the high and moderate/low OP index groups. We have added the following sentence to the Methods section: “In addition, we evaluated the extent to which OPLL occupied the spinal canal in the cervical region and classified the most compressed segment according to the canal narrowing ratio (CNR) [7] as follows: Grade 1, 0%–25%; Grade 2, 26%–50%; Grade 3, 51%–75%; Grade 4, >75%.” We have also added Table 3 and the following sentence to the Results section: “The high OP index group had a significantly higher CNR compared with the moderate/low OP index group”.

Minor comments below

-Why did the authors used a term of “OP-index”? The original report (Kawaguchi et al, Spine, 2013) in which this analysis was used reported this method as “OS-index”.

Response: This index was originally named the “OS index” by Kawaguchi et al. However, we also measured the numbers of ossification of other spinal ligaments, including the OALL, OLF, and nuchal ligaments. To avoid misunderstanding, we defined the numbers of OPLL as the OP index, that of OALL as the OA index, and that of OLF as the OL index in a previous paper.[2] Dr. Kawaguchi has already accepted this amendment.

[2] Mori K., et al. Prevalence and distribution of ossification of the supra/interspinous ligaments in symptomatic patients with cervical ossification of the posterior longitudinal ligament of the spine: a CT-based multicenter cross-sectional study. BMC Musculoskelet Disord. 2016 Dec 1;17(1):492. doi: 10.1186/s12891-016-1350-y.

-The authors stated that “The OP-index value was markedly higher, especially in the thoracic spine, in the high OP index group.” However, it looks like that there is no data which suggests this idea. P value cannot be the evidence for this statement.

Response: The number of ossifications in the thoracic spine shown in Figure 2A can explain the result. To clarify this point, we have revised the text as follows: “The number of ossifications, especially in the thoracic spine, was markedly higher in the high OP index group.” (page6, lines192-193)

-In the discussion section, the authors stated that cervical OP-index, female sex, and obesity were significantly correlated with OP-index. Which data indicated the significant correlation between cervical OP-index or obesity with OP-index? I cannot find them.

Response: We apologize for this error. This evidence was investigated in a previous paper.[3] Therefore, we have revised the text as follows: “A previous report showed that the amount of OPLL (i.e., the OP index) was significantly correlated with the cervical OP index, female sex, and obesity.” (page8, lines227-228)

[3] Hirai, T.; Yoshii, T.; Iwanami, A.; Takeuchi, K.; Mori, K.; Yamada, T.; Wada, K.; Koda, M.; Matsuyama, Y.; Takeshita, K.; et al. Prevalence and Distribution of Ossified Lesions in the Whole Spine of Patients with Cervical Ossification of the Posterior Longitudinal Ligament A Multicenter Study (JOSL CT study). PLoS One 2016, 11, e0160117,

Round 2

Reviewer 2 Report

The authors appropriately answered to my concerns. Now, I can recommend this article for publication.